# Tuning the Morphology and Gas Separation Properties of Polysulfone Membranes

**DOI:** 10.3390/membranes12070654

**Published:** 2022-06-25

**Authors:** Steven Kluge, Tillmann Kose, Murat Tutuş

**Affiliations:** Fraunhofer Institute for Applied Polymer Research IAP, 14476 Potsdam, Germany; tillmann.kose@iap.fraunhofer.de (T.K.); murat.tutus@iap.fraunhofer.de (M.T.)

**Keywords:** biogas, gas separation, polysulfone membrane, membrane morphology, flat sheet membranes

## Abstract

The present work deals with the modification of casting solutions for polysulfone gas separation membranes fabricated by wet-phase inversion. The aim was to fabricate membranes with thin gas separation layers below one micrometer of thickness and a sponge-like support structure. With decreasing thicknesses of the separation layers, increasing permselectivities were observed. For the first time, we could show that permeabilities and diffusion coefficients of certain gases are orders of magnitude lower in separation layers of membranes below 500 Å of thickness compared to separation layers with a thickness above 1 micrometer. These results indicate that the selection of the solvent system has a huge impact on the membrane properties and that the permeability and diffusion coefficient are not material-related properties. Thus, they cannot be applied as specific indicators for gas-separating polymers. In this publication, scanning electron microscopy and gas permeation measurements were carried out to prove the gas separation properties and morphologies of polysulfone membranes.

## 1. Introduction

A circular economy is currently being addressed more and more frequently in industry, according to which waste should serve as educts for other processes. In order to do justice to a circular economy, it is often necessary to process waste streams. Exhaust gases, for example, often contain valuable components that can be recovered with the help of membranes in order to return them to the original process or to provide them as educts for other processes; carbon dioxide (CO_2_) can be separated or hydrogen (H_2_) can be recovered [1].

Another application for membrane gas separation is the growing biogas sector. In a biogas reactor, gases are generated which consist of up to 95% CO_2_ and methane (CH_4_) [2]. If CH_4_ is to be used in combined heat and power plants to generate energy or to be fed into the existing natural gas network, it must first be separated from CO_2_ and other trace gases [3]. Previous processes such as pressure swing adsorption, amine washing or cryogenic distillation require a high regulatory effort and high energy costs to feed CH_4_ of the required purity (at least 98%) into the natural gas network. Separating gas with membranes is a simpler and more environmentally sustainable variant in terms of plant technology as chemicals for purification can thus be dispensed with [2,3,4].

The aforementioned applications are already partially executed with dense polymer membranes. Gas separation via dense membranes is based on the different permeation behavior of gases through a membrane. Permeation consists of two steps: The first step is the sorption of gases in the membrane material [5]. In a second step, dissolved gases are able to diffuse due to an external driving force [6,7,8,9]. The driving force is a difference in the chemical potential between the feed and the permeate side of a membrane and is often applied by way of a pressure difference. Combining sorption and diffusion results in the permeance of a molecule through a membrane. Hence, permeation consists of solution and diffusion and can be expressed as follows:Permeability (P) = Solution (S) × Diffusion (D)(1)

In addition to polysulfone (PSU), cellulose acetate (CA) and polyimides (PI) are often used for gas separation. CA and PI usually exhibit higher selectivity for the gas pair CO_2_/CH_4_ as well as higher CO_2_-permeabilities compared to PSU. The latter, on the other hand, exhibits a higher plasticization pressure for CO_2_ [10]. At the plasticization pressure, a given gas leads to a large swelling of the membrane. This swelling causes strongly increasing permeabilities of all gases in a mixture, resulting in strongly decreasing selectivities [10,11]. The plasticization pressure for CO_2_ is 10 and 12 bar for CA and PI, respectively. Under the same conditions, PSU shows a plasticization pressure above 34 bar [10,12,13]. In gas separation applications relevant for industry, this high CO_2_ plasticization pressure of PSU is of special interest [14].

In literature, a number of fabrication techniques for PSU flat sheet membranes are described, such as non-solvent induced phase separation (NIPS), evaporation-induced phase separation (EIPS) or temperature-induced phase separation (TIPS) [15]. The non-solvent induced phase separation in particular has been researched thoroughly [16,17,18,19,20,21,22,23,24,25,26,27]. Two methods within this technique have been developed: the wet-phase inversion method and the dry/wet-phase inversion method. In the wet-phase inversion method, the polymer film is precipitated in a non-solvent directly after film casting. Dry/wet-phase inversion describes a method which includes a determined evaporation step after the film casting but before the precipitation in a non-solvent [16,18,28,29].

Pfromm et al. found that membrane morphology and separation characteristics can be set by changing the method of membrane formation [8]. Moreover, the membrane’s porosity or density can be chosen. Parameters for tuning membrane morphology comprise, amongst others, the composition of the casting solution, the precipitation medium, the temperature of the precipitation bath, the residence times, the support material and the post-treatment [15,30,31,32]. To the best of our knowledge, there is no model available to predict membrane morphology by simulations. All predictions are based on assumptions and practical experiences, which are in turn based on empirical and physicochemical data [30]. Predictions of expected membrane morphology by means of simulations are a focal point of current research [33].

Based on a polymer dissolved in an organic solvent and water as a precipitation medium, membrane morphology can be estimated by solubility parameters of solvents, non-solvents and polymers [23]. Water is often used as a precipitation medium as most membrane polymers are soluble in organic solvents but not in water. For the precipitation in water, the organic solvent needs to be completely miscible with water. Compared to organic molecules, the water molecule is polar and smaller with respect to the kinetic diameter. This results in a diffusion coefficient of water diffusing into the wet film one to two orders of magnitude above the diffusion coefficient of organic solvents diffusing out of the wet film whilst precipitating [30].

Theoretically, the osmotic pressure of water can reach 100 bar [30]. The polymer solution is displaced accordingly in the direction of penetration by the penetrating water before it is solidified by the solvent exchange [21,34]. Therefore, membranes with a thin separation layer and large macrovoids are obtained mostly [30]. This kind of membrane shows high gas fluxes but low mechanical stability. The time at which solidification occurs is therefore crucial for membrane morphology as well as gas separation properties. By changing various parameters such as polymer concentration or the solvent system, or adding additives such as cross-linkers, it is possible to influence membrane morphology [30,35]. All of the aforementioned changes can slow down solvent exchange mainly by increasing the casting solution’s viscosity, which leads to thicker separation layers with a sponge-like support structure as void formation is suppressed [15,30]. This kind of membrane exhibits low gas fluxes but high mechanical stability.

Another crucial characteristic for gas separation is the free volume of a polymeric membrane. The free volume of a polymer describes the free space between polymer chains. It is often expressed by means of the measurable value of the specific volume. The specific volume V_sp_ of a polymer consists of the occupied volume V_oc_ and the unoccupied or free volume V_f_ [36]:V_sp_ = V_oc_ + V_f_(2)

Occupied volume refers to the polymer chains in the membrane, whereas free volume refers to the space between the polymer chains. Hence, the free volume has an influence on diffusion and permeation in a polymeric membrane [37]. Yet, Kesting found a larger influence of the size of solvent molecules than the quality of the solvents on the free volume in a polymeric membrane [38]. Ismail et al. proved a correlation between shear rate during the casting of the polymer film and gas separation properties of the resulting membrane [24]; the higher the shear rate, the better the gas separation properties. Ismail et al. assumed that the molecular orientation of the polymer chains is affected by the shear rate. The authors summed up that a higher shear rate stretches the polymer chains, which leads to more tightly packed chains in the separation layer of asymmetric membranes. A tighter packaging implies a lower free volume in the polymeric membrane and therefore higher selectivities [24]. With a lower free volume, selectivities increase because diffusion coefficients of larger molecules are affected to a greater degree by a loss of free volume than the diffusion coefficients of smaller ones. Pinnau et al. found higher selectivities for asymmetric membranes compared to bulky homogeneous membranes as well [16].

According to the literature mentioned so far, the aim of the present work is to develop thin separation layers with a sponge-like support structure as membrane morphology via wet-phase inversion. To achieve this morphology, NIPS, especially wet-phase inversion, is to be utilized since EIPS as a fabrication method is often too slow for industrial membrane fabrication because of the low-volatile solvents used in the process. Additionally, a goal of this research is to verify that the permeability and diffusion coefficient of a certain gas are not material-related properties. They depend on the composition of the casting solution, amongst other factors.

## 2. Experimental

### 2.1. Materials

Polysulfone, P-3500 LCD MB7 (78.6 kg/mol, Solvay), was used to fabricate PSU flat sheet membranes. Polydimethylsiloxane (Elastosil RT 625 A and B, PDMS, Wacker Chemie AG, Munich, Germany) was used to coat the fabricated polysulfone membranes. The solvents tetrahydrofuran (99.9%; Roth), *N*,*N*-dimethyl acetamide (99.5%; Roth), methanol (99.9%; Roth), ethanol (99.5%; Roth) and n-heptane (≥95%; Roth) were used as received. 

### 2.2. Membrane Preparation

The desired amount of solvents was filled into a capped flask and PSU pellets were added. Polysulfone and solvents were mixed in the closed flask for 16 h on a roller bench until a homogeneous, yellowish polymer solution was obtained. In order to shift the solubility equilibrium closer to the demixing point, a certain mass of methanol was then added to the polymer solution drop by drop and further mixed until the solution was homogeneous again. The obtained casting solution was then poured onto a glass plate and fabricated into a film with the aid of a casting knife set to 250 µm at a speed of 17 mm/s (1 m/s). Immediately after casting, the glass plate was immersed in the precipitation bath (distilled water or ethanol) for 4 min. To wash away the remaining solvents, the precipitated membranes were rinsed with distilled water for about 1 min on each side. Afterward, the washed membranes were left to dry at room temperature in a dust-free box for at least 16 h. After they had dried, the membranes were coated with PDMS (9:1 = component A:component B, 70% by mass in n-heptane). The PDMS film was cross-linked for 1 h at 150 °C in a drying oven.

### 2.3. Scanning Electron Microscopy

To examine the membrane structures obtained, scanning electron micrographs were taken with a GeminiSEM 300 (Carl Zeiss NTS, Oberkochen, Germany). For this purpose, the samples were immersed in liquid nitrogen, broken and then applied to a conductive carrier before they were vapor-deposited with a 4 nm thick layer of platinum. The samples were measured at an acceleration voltage of 5 kV. The SEM images in this paper only include membrane samples from polysulfone membranes without a PDMS protective layer, since the focus of this work is on PSU flat sheet membranes.

### 2.4. Gas Permeation Measurements

Gas permeation tests were conducted using a time-lag apparatus. Permeabilities of the coated membranes for hydrogen (H_2_), oxygen (O_2_), nitrogen (N_2_), methane (CH_4_) and carbon dioxide (CO_2_) were determined using the time-lag method at 30 °C [39]. The measurements were performed after an evacuation duration of 1.5 h after every sample exchange and of 12 time lags (Θ) between the measurements (minimum 3 min, maximum 6 h). To minimize measurement uncertainties, each gas was measured twice and at different feed pressures of 390 mmHg (520 mbar) and 600 mmHg (800 mbar). The permeate pressure was set to 7.5 × 10^−5^ mmHg (1 × 10^−4^ mbar) at the beginning of each measurement. To avoid solvent residues in the membrane influencing the gas permeation measurement, a manual step was added: the vent to the turbo pump was closed for 3 min, as was the vent to feed volume directly above the membrane cell. In this setup, the volume in which pressure increase can happen is minimized. If the pressure was raised not more than 0.2 mmHg (0.3 mbar) in a time span of 3 min, the samples were assumed to be solvent-free and the gas permeation measurements were started.

The apparatus measures the time-dependent pressure increase of the permeate in order to obtain a time–pressure curve for every gas. A schematic representation of such a curve is given in Figure 1. 

From the time-lag measurements, two properties can be determined: the permeability $P_{i}$ and the diffusion coefficient $D_{i}$ of gas i. The gas permeability $P_{i}$ is calculated as follows:(3)$$P=\frac{l\cdot V\left(\mathrm{STP}\right)}{A\cdot t\cdot \left(\Delta p_{F}-\Delta p_{P}\right)}$$
where l is the membrane thickness (cm), V(STP) is the volume of the permeate (cm^3^), A is the effective membrane area (cm^2^), t the passed time within the quasi-stationary range (s), $\Delta p_{F}$ is the difference in the feed pressure within the quasi-stationary range (cmHg) and $\Delta p_{P}$ the difference in the permeate pressure within the quasi-stationary range (cmHg). Permeabilities are given in Barrer (1 Barrer = 10^−10^ $\frac{{\mathrm{cm}}^{3}\left(\mathrm{STP}\right)\cdot \mathrm{cm}}{{\mathrm{cm}}^{2}\cdot s\cdot \mathrm{cmHg}}$).

Diffusivities $D_{i}$ of a certain gas i can be calculated from the time lag Θ according to the following equation:(4)$$D=\frac{l^{2}}{6\Theta }$$

The time lag Θ is determined graphically from the time–pressure diagrams by extrapolating the stationary curve to the time axis (Figure 1).

The permeability selectivity (permselectivity) $\alpha_{A,B}$ of a membrane in relation to two gases *A* and *B* is also called ideal selectivity and is determined according to Equation (5) from the ratio of the permeabilities of gases *A* and *B* within the linear range of the pressure increase.
(5)$$\alpha_{A,B}=\frac{P_{A}}{P_{B}}$$

Another measure of a membrane’s performance is the ability to separate a gas mixture. In this case, a separation factor is obtained according to Equation (6), where *x_i_* is the concentration of gas *i* in the feed and *y_i_* is the concentration of gas *i* in the permeate.
(6)$$\alpha_{A,B}^{*}=\frac{y_{A}/y_{B}}{x_{A}/x_{B}}$$

The separation factor typically exhibits lower values than the permselectivity due to effects such as plasticization of the membrane material caused by molecules such as water and CO_2_ [10,40]. With the help of the gas permeation setup at Fraunhofer IAP, it is possible to determine permselectivities.

The solubility can be calculated as the ratio of permeability and diffusivity according to the following equation:(7)$$S=\frac{P}{D}$$

In the present work, solubility has no independent significance and is only mentioned demonstratively. 

## 3. Results and Discussion

The membrane morphology of PSU membranes we reported elsewhere revealed a thick gas separation layer upon a sponge-like support layer [29]. Due to the high resistance of a thick separation layer, a separation layer as thin as possible is desirable. The latter leads to higher gas fluxes through the membrane while the gas separation properties are expected to remain the same.

### 3.1. Scanning Electron Microscopy

Membranes of the aforementioned publication featured two layers: a gas separation layer and a support structure. The separation layer of the membranes could be observed in a range between 3.5 and 11 µm. Typically, gas separation membrane layers of this thickness demonstrate a high resistance to gas transport leading to a low gas flux. Therefore, a much thinner gas separation layer is needed. The sponge-like support structure of those membranes, however, results in mechanically stable membranes. The support structure displayed a thickness of around 23 µm for membranes precipitated in ethanol and a thickness of 28 to 30 µm for membranes precipitated in distilled water. Scanning electron microscope (SEM) images of membranes with this type of morphology are depicted in Figure 2 for comparison.

The aim of the current paper was therefore to fabricate a gas separation layer as thin as possible with a sponge-like support structure, which grants sufficient mechanical stability to the membrane by means of wet-phase inversion. Since gas separation is a pressure-driven process with feed pressures up to 100 bar, mechanical stability should be given [41,42,43]. The first step towards a thinner separation layer was to lower the polymer concentration in the casting solution from 20 to 15 wt.-%.

This resulted in a separation layer with an average thickness of 2 to 3 µm (see Figure 3). Under the same fabrication conditions, a 20 wt.-% casting solution resulted in an average separation layer thickness of 6 µm (see Figure 2). Thus, by lowering the polymer concentration from 20 wt.-% to 15 wt.-%, the separation layer could be decreased by 50 to 67%.

Nevertheless, the support structure of the membrane showed a sponge-like structure with larger voids. This can be explained by the lower polymer concentration of the casting solution. After hand casting, the resulting wet film has a larger volume of a polymer-poor phase while precipitation is occurring. This leads to fewer but larger voids in the sponge-like support structure. The thickness of the support structure could be determined to be around 22 to 25 µm, while with the 20 wt.-% casting solutions, the thickness was gauged in the range of 23 to 30 µm. We expected this decrease in the separation and the support structures since a lower polymer concentration in the casting solution means less dissolved material in the same volume of casting solution compared to a higher polymer concentration.

The desired thickness of the separation layer of several hundreds of nanometers, however, could not be achieved with this formulation. Lowering the polymer concentration to even smaller amounts showed a larger number of defects in the separation layer of the resulting membranes or even porous membranes. Therefore, we decided to change the overall composition of the casting solution. The composition of the casting solution can influence the evaporation rate of solvents before precipitation and the speed of solvent exchange during precipitation. Both steps, the evaporation of solvents before precipitation and the velocity of solvent exchange during precipitation, are crucial for the resulting membrane morphology. 

Fast solvent evaporation before precipitation often results in a gel layer on top of the membrane. This layer occurs because the polymer is enriched on the boundary between the wet polymer film and air. This polymer-rich phase reduces the velocity of solvent exchange during precipitation. A decreased rate of solvent exchange occurs because the high viscosity of the gel layer lowers the diffusivities of the precipitation medium and the solvents. Emerging membranes therefore feature dense gas separation layers. 

However, the gel layer of the wet film not only affects the velocity of solvent exchange. The solvents themselves show different interactions with the polymer, with other solvents or non-solvents in the casting solution and with the precipitation medium. This suggests that adding another solvent can change the solubility of the polymer and the viscosity of the casting solution, even if all other parameters, e.g., polymer concentration, are kept constant. Using an additional solvent can suppress or promote the appearance of voids in the support structure of a membrane, which is a result of different velocities of solvent exchange during precipitation or different viscosities of the casting solution. This also implies that the ratio of solvents used for the casting solution results in a critical point when it comes to membrane morphology.

We denote that significant parameters such as the temperature of fabrication, temperature of the precipitation bath, room humidity and gap height of the casting knife, amongst others, were not included in the research for this publication. These factors, however, influence the resulting membrane morphology as well.

The hypothesis for achieving a thin separation layer with a sponge-like support structure was to lower the rate of evaporation before precipitation and simultaneously keep the velocity of solvent exchange low. To lower the rate of solvent evaporation, we added dimethyl acetamide (DMAc) to the casting solution of 20 wt.-% PSU. The composition of the casting solution and the casting conditions are specified below.

The concentration of MeOH in the casting solution had to be lowered due to precipitation of PSU when an amount of 3.25 g (13 wt.-%) MeOH was added like in the formulations without DMAc. First trials with water as a precipitation medium resulted in membranes with a very thin separation layer. Defect-free membranes were only insufficiently reproducible. The separation layer was often defected by pinholes leading to membranes with almost no permselectivity for different gas pairs. Pinholes occur when solvents evaporate too slowly while DMAc is used additively. Therefore, the gel layer on top of the wet film before precipitation turns out very thin. As a result, water from the precipitation bath can penetrate this layer to form pinholes.

We assumed that lowering the velocity of precipitation would lead to thicker separation layers, as the velocity of solvent exchange is lower when ethanol instead of water is used as precipitation medium. The reason for this is the sizes of the molecules of water and ethanol: Larger molecules show lower diffusivities, which implies a lower velocity of solvent exchange. In this case, ethanol (4.2 Å) has a larger kinetic diameter than water (2.6 Å) [14]. The morphology of the fabricated membrane of the composition in Table 1 and with ethanol used as precipitation medium is demonstrated in Figure 4.

With the composition of the casting solution shown in Table 1 and ethanol as precipitation medium, we were able to achieve the same membrane morphology as with the former formulation that uses only THF as solvent and MeOH as cosolvent. A two-layered membrane morphology consisting of a dense separation layer and a sponge-like support structure was observed. The separation layer showed a thickness varying from 9 to 12 µm, and the support structure exhibited a consistent thickness of 75 µm. Since the membrane morphology matches the morphology of our previous publication, the separation layer showed large expansion.

The change from water to ethanol as a precipitation medium led from very thin and defective separation layers to layers that were too thick for gas separation applications. This implied that the casting solution needed to be tuned for the fabrication of a very thin but defect-free separation layer and a sponge-like support structure. 

In a first attempt, we kept the solvent ratio and the amount of MeOH in the solution constant but increased the polymer concentration to 25 wt.-%. The precipitation medium used was water. It was to be expected that the higher polymer concentration would prevent the penetration of water through the wet casting film during precipitation. This is due to the higher viscosity of the casting solution as a higher amount of polymer is dissolved. Thereby, the diffusivity of the precipitation medium is lowered. Consequently, the penetration of the emerging separation layer is stopped [15,22,30]. The resulting membrane morphology for the more highly concentrated casting solution is shown in Figure 5.

The thickness of the separation layer was lowered drastically compared to the membranes from 20 wt.-% PSU casting solutions precipitated in ethanol. In contrast to membranes fabricated from a 20 wt.-% casting solution, membranes from a 25 wt.-% PSU casting solution were widely free of defects, and large drop-shaped voids were formed. These voids can cause mechanical instability in membranes used for high-pressure applications. Hence, void-filled membranes can be damaged during gas separation, which needs to be prevented. 

In order to fabricate membranes with a sponge-like support structure but a thin separation layer, we increased the polymer concentration to 30 wt.-%. Since these casting solutions exhibited a higher viscosity, the formation of voids due to penetrating water during precipitation should have been eliminated. Figure 6 shows the result for a membrane made from a casting solution consisting of 30 wt.-% polysulfone in the given solvent system and precipitated in distilled water.

It can be noted that in general the same morphology occurred with the 30 wt.-% solution as it did with the 25 wt.-% solution: Membranes showed a very thin separation layer with a support structure filled with drop-shaped voids. As with the 25 wt.-% casting solutions, the separation layer was widely free of defects. SEM images revealed a thickness of the separation layer significantly smaller than 1 µm. Judging from those images, we assumed a thickness of several dozens to hundreds of nanometers for the separation layer. 

Nevertheless, the desired membrane morphology had not yet been achieved. This is why we decided to tune the ratio of THF to DMAc. Simultaneously, the concentrations of PSU and MeOH were raised in comparison to the composition in Table 1. The increase in PSU and MeOH in the casting solution was meant to prevent the formation of voids within the support structure. The higher viscosity due to the higher polymer content in the casting solution decreases the diffusivity of water from the precipitation bath into the wet polymer film during precipitation. MeOH, which diffuses out into the precipitation medium during precipitation and thus in the opposite direction of the diffusing water from the precipitation bath, also decreases the velocity of precipitation. 

Both changes were expected to suppress the formation of voids. Additionally, the use of more THF in the casting solution was expected to form a sponge-like support structure since casting solutions with THF as the sole solvent resulted in membranes without drop-shaped voids. The parameters for these new casting solutions read as follows:

These changes were based on distilled water as precipitation medium. Using water for the production of membranes is more desirable as it is cheaper than ethanol. Moreover, it is easier to handle due to its lower volatility, and no recovery is needed compared to ethanol. In addition, ethanol vapors have to be collected because of environmental restrictions, whereas water vapor is allowed to enter the atmosphere unhindered. 

The resulting morphology of membranes prepared from the composition and conditions described in Table 2 is shown in Figure 7.

The membrane in Figure 7 exhibits a very thin separation layer upon a sponge-like support structure. With the help of SEM images, we estimated that the separation layer had a thickness of several hundreds of nanometers. This expansion of the separation layer was desired as it is in an interesting region for gas separation applications. The separation layer showed no defects via SEM observations. Furthermore, the support structure should provide mechanical stability due to the sponge-like structure [36]. In the left part of Figure 7, drop-shaped voids can still be identified just below the separation layer. In addition, the growth of these voids was successfully suppressed, and their dimensions remained low compared to voids in Figure 5 and Figure 6. The voids’ impairment of the mechanical properties of the membranes is therefore neglected.

To this point, it was possible to fabricate membranes with a thin separation layer and a sponge-like support structure without obvious defects in the separation layer via the wet-phase inversion process. Then, the gas separation properties of the membranes were characterized.

### 3.2. Gas Permeation Measurements

In order to characterize the gas separation properties, single gas permeation measurements were conducted. Gases tested were O_2_, N_2_, CH_4_ and CO_2_. To create a benchmark, a symmetric PSU membrane without support structure was prepared additionally via EIPS. Since this membrane has no voids and consists just of one dense layer, we suggested having no pinholes through the whole membrane layer. Therefore, this membrane was expected to indicate the maximum permselectivities possible. Hence, lower permselectivities from membranes prepared via wet-phase inversion can be a hint for not completely defect-free separation layers. Even pinholes in the sub-micrometer scale can drastically reduce the gas separation properties of a membrane, as proposed by Henis and Tripodi [44]. Table 3 summarizes the data obtained from PDMS-coated PSU membranes.

As expected, the PSU membranes prepared via EIPS (PSU30_EIPS) showed the highest permselectivities except for the gas pair CO_2_/N_2_. The permselectivity is the ratio of permeabilities (see Equation (7)). If only slight differences in permeabilities of N_2_ or CH_4_ occur, the permselectivity changes significantly, due to the low values of N_2_- and CH_4_-permeabilities. Almuhtaseb et al. showed even higher selectivities for PSU membranes prepared via EIPS for the gas pair CO_2_/CH_4_ [45]. In their study, other solvent systems, e.g., THF and chloroform, were used, and the characterization method differs from that of this study. Therefore, selectivities in contrast to permselectivities were obtained. Furthermore, measurements of Almuhtaseb et al. were conducted at 20 °C whereas our membranes were tested at 30 °C. At lower temperatures, membranes are known to show better separation performance.

**Table 3 membranes-12-00654-t003:** Permeabilities and permselectivities of PSU and PDMS flat sheet membranes. * data given in GPU (=gas permeation unit, 1 GPU = 1 × 10^−6^ cm^3^ (STP) (cm^3^·s·cmHg)^−1^) [46].

Membrane	Permeability (Barrer)				Permselectivity (–)			References
	O_2_	N_2_	CH_4_	CO_2_	O_2_/N_2_	CO_2_/N_2_	CO_2_/CH_4_	[29]
PSU20	1.05	0.20	0.21	6.18	5.4	30.9	29.3	[29]
PSU15	1.06	0.18	0.20	6.01	5.8	32.7	30.4	this work
PSU20_EtOH	1.99	0.33	0.38	11.01	6.3	34.7	31.4	this work
PSU25_80:20	2.28	0.37	0.40	12.98	6.1	34.8	32.4	this work
PSU30_80:20	2.18	0.35	0.36	11.71	6.3	33.9	32.6	this work
PSU25_94:6	1.98	0.29	0.29	10.84	6.9	37.5	37.2	this work
PSU30_EIPS	1.73	0.25	0.23	8.95	7.1	36.5	38.5	this work
PSU (EIPS)	---	---	0.65	30.0	---	---	50	[45]
PSU (EIPS)	---	---	0.72	24.8	---	---	35	[45]
PSU	1.4	0.25	0.25	5.48	5.6	21.9	21.9	[47]
PSU	8.83 *	2.26 *	2.84 *	27.77 *	4.1	12.3	9.9	[24]
PDMS (coating layer)	570	266	860	2770	2.1	10.3	3.2	this work
PDMS	500	250	800	2700	2.0	10.8	3.4	[47]

The measured O_2_-permeabilities of asymmetric membranes of this study lie in the range between 1.05 and 2.28 Barrer, which is in good agreement with data from the literature. N_2_ (0.18–0.37 Barrer) and CH_4_ (0.20–0.40 Barrer) exhibit permeabilities that are also in good agreement with literature data, whereas CO_2_-permeabilities (6.01–12.98 Barrer) revealed a slightly increased permeability in comparison with data from the literature. These numbers result in permselectivities as high as those in the literature or even higher [16,17,25,28,47,48]. In particular, the membranes PSU25_94:6 and PSU30_EIPS show significantly higher permselectivities for the gas pairs O_2_/N_2_ and CO_2_/CH_4_. With the PSU30_EIPS membranes, a permselectivity of 7.1 was reached for the gas pair O_2_/N_2_. PSU25_94:6 membranes showed just a slightly decreased permselectivity of 6.9. Hence, these membranes were mostly defect-free. Referring to Pinnau and Koros, the separation layers of asymmetric membranes can be considered “essentially defect-free” if the O_2_/N_2_ permselectivity is within 85% of the permselectivity of a symmetric membrane without substructure [16]. This applies to all asymmetric membranes in this study. Even pores in the range of 5 to 10 Å over a small area fraction will drastically decrease the separation properties of a membrane [44]. To avoid even those small pores, all asymmetric membranes were coated with a PDMS protective layer. Permselectivities imply that the PDMS protective layer has no influence on gas transport in the multilayer system consisting of PSU and PDMS. The influence of the protective layer on gas transport in a multilayer system is generally negligible if [16]:The number of defects in the separation layer of asymmetric membranes is low (<10^−6^);The coating material shows significantly higher gas permeabilities compared to the coated material (see Table 3);The thickness of the protective layer is minimized.

That means that any differences in permeability or diffusion coefficients are due to the characteristics of the PSU membranes. On the other hand, the uncertainty of measurement plays also an important role in changes in the permselectivity, especially when permeabilities of N_2_ and CH_4_ were used for the calculation of permselectivities.

Pinnau and Koros showed permselectivities for O_2_/N_2_ of PSU flat sheet membranes not above 6.7, even though these PSU membranes were coated with PDMS [16]. Furthermore, Pinnau and Koros assumed that the molecular orientation of PSU in very thin separation layers differs from that in the bulk polymer. This assumption was confirmed later [24,47]. In contrast, we showed very similar results for membranes prepared by wet-phase inversion and membranes prepared via EIPS. We therefore hypothesize that molecular orientation is not just a result of the phase inversion method. It is also a result of the composition of the casting solution. Some indication of this hypothesis was already given in the research of Kesting [38].

PSU membranes of this paper fabricated under different conditions showed permselectivities for O_2_/N_2_ typical for PSU flat sheet membranes. Permselectivities of CO_2_/CH_4_, however, showed higher values compared to membranes fabricated under the same conditions in the literature. Literature values ranged below the membranes PSU25_94:6 and PSU30_EIPS [16,18,24,25,28]. This trend is a clear indication that just by tuning the composition of casting solutions and by varying the casting conditions, the separation performance of flat sheet membranes can be optimized significantly. Compared to the PSU20 membranes, an increase in the permselectivities of 28% for O_2_/N_2_, 21% for CO_2_/N_2_ and 27% for CO_2_/CH_4_ was achieved with the PSU25_94:6 membranes.

Our explanation for this purpose is as follows: referring to Kesting’s work and the theory of polymer solutions, we conclude that the molecular orientation of PSU is different from that of other casting solutions because of the choice of solvents and cosolvents. Different solvents and solvent systems lead to different conformations of the polymer chains before and during precipitation. This can be justified by the solubility parameters of all interacting molecules. A two-component system, consisting of a solvent and a polymer, can reach a theta state, which means that the polymer in solution shows a so-called ideal chain. In the theta state, there are balanced interactions between solvent and monomers, meaning that there is no swelling (strong interaction between solvent and monomer) and no chain collapse or clustering (poor interaction between solvent and monomer) of the polymer chains. These balanced interactions lead to an unperturbed or ideal chain. Therefore, this kind of solvent is called a theta solvent. For this two-component system, the interaction parameter $\chi_{1,2}$ can be estimated by Equation (8) as follows:(8)$$\chi_{1,2}=\frac{V_{1}}{RT}{\left(\delta_{1}-\delta_{2}\right)}^{2}$$
where *V*_1_ is the volume of the mixture, *R* is the ideal gas constant, *T* is the absolute temperature and δ is the solubility parameter of the solvent (1) and the polymer (2).

The stronger the interaction between solvent molecules and monomers of the polymer chains, the wider the polymer chains are expanded. In this case, a good solvent leads to a swollen polymer chain in solution, which results in a membrane with a larger free volume. In contrast, a stronger interaction between polymer chains in solution gives collapsed chains. In this case, a poor solvent leads to chain collapse or clustering, which results in a lower free volume in the resulting membrane.

In addition, the interaction between solvent and cosolvent molecules plays a role in determining the expansion of polymer chains in our casting solutions. This implies that Equation (8) is influenced by more interactions between all contained molecules. A denser packaging of the polymer chains is possible, but also there can be a higher free volume in case of strong interaction between solvent molecules and polymer chains. On the macroscopic scale, we see a different behavior of the gas permeation properties of the membranes and therefore different gas separation properties.

With the aid of Equation (9), the layer thickness of the asymmetric membranes can be calculated according to the obtained gas fluxes from the gas permeation experiments [16]:(9)$$L=\frac{P_{i}}{{\left(P/L\right)}_{i}}$$
where *P_i_* is the permeability of gas i, determined in the symmetric PSU membrane of known thickness, and (*P*/*L*)*_i_* is the pressure-normalized gas flux of the asymmetric membranes. Equation (9) assumes that the porous substructure of the support structure has no resistance to gas transport within asymmetric membranes. A measured thickness of 74.6 µm for PSU30_EIPS and N_2_ as gas i give the thicknesses for the separation layers of asymmetric membranes as seen in Table 4 when Equation (9) is applied.

The calculated thicknesses of separation layers in the asymmetric PSU membranes are in good agreement with the estimations from SEM images shown in Section 3.1. Nonetheless, the decreased thickness of the separation layers leads to much lower permeabilities for asymmetric membranes when the separation layer thickness of Table 4 is applied to calculate permeabilities. With the example of CO_2_, it is clear that permeabilities are two to three orders of magnitude lower compared to data obtained before. Permeabilities in Table 3 were calculated according to the permeability of the inert gas helium (He). The permeability for He was chosen to be 20.2 Barrers because this permeability was achieved by a membrane with a known thickness (PSU30_EIPS). Since the permeability is expected to be a material-related property and the composition of the casting solutions was not changed significantly, it was straightforward to calibrate the thickness of the separation layer in asymmetric membranes to a He permeability of 20.2 Barrers. The composition of the casting solution, i.e., change of solvents and cosolvents, is known to change the permeabilities of gases in one and the same membrane material [13,17,28,43,46,49,50,51]. He was chosen because the solubility of He in PSU is low compared to the other gases tested, and therefore the assumption was that no swelling of the PSU membranes influences the permeability data obtained with He. In this method, the thicknesses of asymmetric membranes need to be set to 19.3 µm (PSU25_80:20), 29.9 µm (PSU30_80:20) and 30.2 µm (PSU25_94:6), which are obviously much higher than the calculated values in Table 4. Since the thickness of the PSU membranes PSU30_EIPS is known and the pressure-normalized gas flux of asymmetric membranes is independent of the thickness of a membrane, we consider a higher reliability for the data calculated in Table 4. This implies very low permeabilities for all measured gases and is in agreement with previous conclusions of Pfromm et al., Pinnau and Koros and Aitken et al. [8,17,18,52]. The theory found in literature is that very thin separation layers contain more tightly packed polymer chains and therefore have differences in free volume, cohesive energy density and probably other factors, resulting in lower permeabilities compared to the bulk polymer.

Shishatskii et al. proved decreasing density in polymer membranes with decreasing film thickness. As a result, diffusion coefficients and permeabilities of the tested gases were decreased [53]. In this study, we could show such a behavior on the macroscopic scale, since the calculated CO_2_-permeabilities are orders of magnitude lower compared to PSU30_EIPS membranes. If the assumed thickness of several dozens to hundreds of nanometers is applied in asymmetric membranes, diffusion coefficients should be orders of magnitude below the values of PSU30_EIPS membranes. Therefore, the diffusion coefficients were calculated with data from Table 3 and Table 4. There is an overview of the diffusion coefficients of O_2_, N_2_, CH_4_ and CO_2_ given in Table 5.

The obtained diffusion coefficients for the measured gases are six orders of magnitude higher in membranes with bulky separation layers compared to membranes with a thin separation layer. Compared to data in the literature, the obtained results for membranes with bulky separation layers are in good agreement [10,13,54,55]. In line with permeability data, the diffusion coefficients of all measured gases are higher in dense PSU30_EIPS membranes compared to all asymmetric membranes with a separation layer below 500 Å [8]. 

The diffusion coefficient shows the ability of movement of a penetrant molecule in a material and is dependent on the penetrant size and the material in which diffusion occurs. In glassy polymers such as polysulfone, diffusion coefficients for gases used to be orders of magnitude lower compared to rubbery polymers such as PDMS, as can be seen in Table 5 [15]. All tested gases in asymmetric membranes showed diffusion coefficients six orders of magnitude lower compared to bulky PSU30_EIPS membranes. However, the obtained diffusion coefficients are in line with the theory of more tightly packed polymer chains, whereas permeabilities showed higher values for asymmetric membranes in Table 3 [8].

Applying Equation (7), lower diffusion coefficients but higher permeabilities (see Table 3) indicate a higher solubility of gases in asymmetric membranes. Related to more tightly packed polymer chains, a higher solubility is not to be expected. With permeability values from Table 5 and diffusion coefficients from Table 4, it is clear that the solubility of gases in asymmetric membranes must be decreased compared to bulky symmetric membranes. Therefore, permeability data in Table 5 are much more reliable, and therefore permeability coefficients are not just material-dependent but also dependent on the layer thickness of the gas separation layer.

Figure 8 depicts the performance of PSU membranes of this work compared with PSU membranes of other studies and other membrane materials.

In gas separation with polymeric membranes, there is a trade-off between permeability and permselectivity, as proposed by Robeson [61]. Figure 8 clearly shows for asymmetric membranes of this study that these membranes show low permeabilities but high permselectivities compared to data for PSU from literature. Moreover, PSU30_EIPS membranes show slightly higher permeabilities and higher permselectivities, except for data from Almuhtaseb et al. As mentioned before, these membranes were measured under different conditions. Generally, we want to note here that no specific operating conditions such as temperature, pressure and conditioning of the samples are considered in the Robeson plot. The operating conditions are, however, very crucial for the performance of a membrane.

These findings imply that especially for thin-film membranes and asymmetric membranes with very thin separation layers, the permeability and diffusion coefficients decrease with decreasing thickness of the separation layer of a membrane due to more tightly packed polymer chains (see Table 4 and Table 5). Additionally, this study shows that the permeability and diffusion coefficients are not applicable as coefficients for certain membrane materials. The permeability and diffusion coefficient, next to other factors such as precipitation medium, strongly depend on the solvents used for preparing the casting solutions. Hence, the normalized gas flux (permeance) should be used for thin-film membranes and asymmetric membranes with very thin separation layers, since the permeance is independent of the thickness of a membrane or separation layer. 

## 4. Conclusions

By changing the solvent system in a first step, we could prepare membranes with the same morphology as with a simpler solvent system. The same morphology was possible when ethanol as a precipitation bath was used instead of distilled water. When using distilled water, asymmetric PSU membranes showed much lower thicknesses of the separation layers but large drop-shaped voids in their support structure, which are expected to lead to less mechanically stable membranes. The separation layers showed thicknesses of 200 to 400 Å compared to around 10 µm for membranes of the same composition but with casting solutions precipitated in ethanol. Through slight changes in the composition of the casting solution, PSU membranes with separation layers below 500 Å and with sponge-like support structures could be fabricated.

Moreover, the normalized gas flux (permeance) has to be taken into account to calculate reliable data for permeabilities of asymmetric membranes. By means of these data, separation layer thicknesses, real permeabilities and diffusion coefficients can be calculated. These findings imply that the permeability and diffusion coefficient of a certain gas are not material-related. We could underline this finding with permeabilities and diffusion coefficients orders of magnitude lower in very thin separation layers with a thickness of several hundreds of Ångströms compared to bulky separation layers with a thickness clearly above 1 micrometer. Based on these values, we could confirm the theory of more tightly packed polymer chains in very thin separation layers of asymmetric membranes. Hence, less free volume leading to a stronger sieving effect can explain increasing permselectivities in asymmetric membranes. Since larger molecules such as N_2_ (3.64 Å) and CH_4_ (3.80 Å) are more affected by a sieving effect of the membrane than smaller molecules such as CO_2_ (3.30 Å) and O_2_ (3.46 Å), the permselectivity increases in very thin separation layers. In line with theory, PSU30_EIPS membranes showed higher diffusion coefficients but also higher permselectivities than asymmetric membranes. PSU30_EIPS membranes were fabricated through evaporation of the solvent system. The results of bulky membranes indicated looser packaging of polymer chains (more free volume) because the membrane formation was slower. This membrane fabrication method resulted in higher diffusion coefficients. The high permselectivities for the gas pairs O_2_/N_2_ and CO_2_/CH_4_ in PSU30_EIPS membranes compared to PSU25_94:6 is in contrast to data from literature and could not be explained finally.

## Figures and Tables

**Figure 1 membranes-12-00654-f001:**
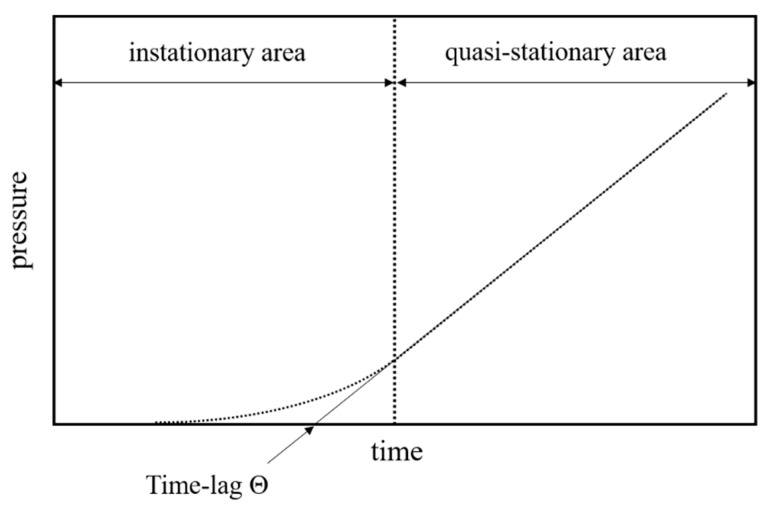
Exemplary depiction of a time–pressure curve obtained from the time-lag measurement.

**Figure 2 membranes-12-00654-f002:**
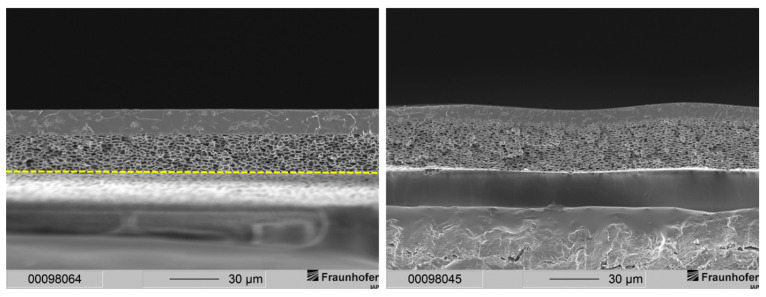
Cross-sections of layered PSU flat sheet membranes without PDMS protective layer. (**Left**): precipitated in ethanol. (**Right**): precipitated in distilled water [29]. The dashed line marks the bottom end of the cross-section of the membrane.

**Figure 3 membranes-12-00654-f003:**
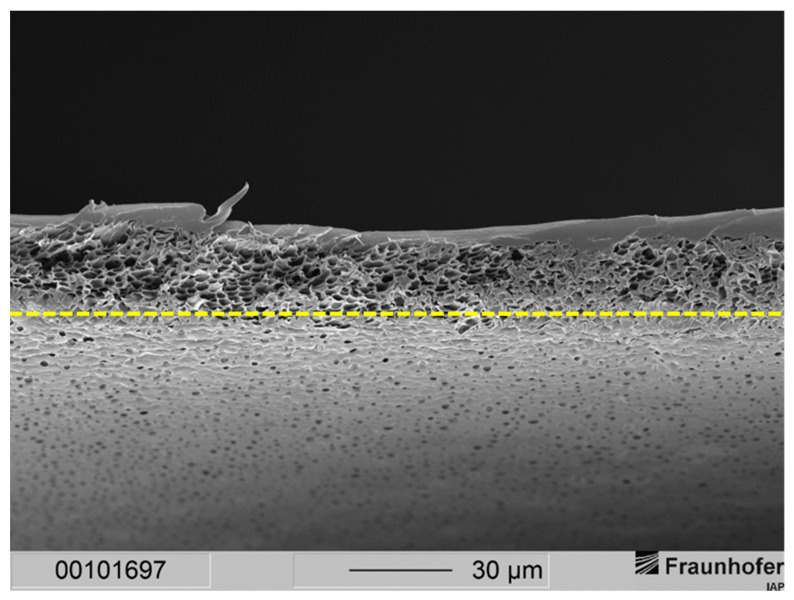
Cross-section of a layered PSU flat sheet membrane cast from a casting solution with 15 wt.-% PSU without PDMS protective layer. The precipitation medium used was distilled water. The dashed line marks the bottom end of the cross-section of the membrane.

**Figure 4 membranes-12-00654-f004:**
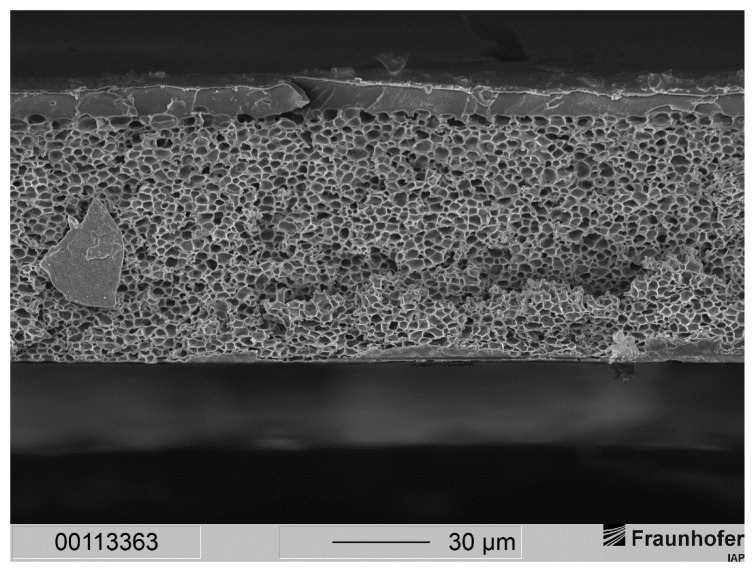
Cross-section of a layered PSU flat sheet membrane fabricated after conditions shown in Table 1.

**Figure 5 membranes-12-00654-f005:**
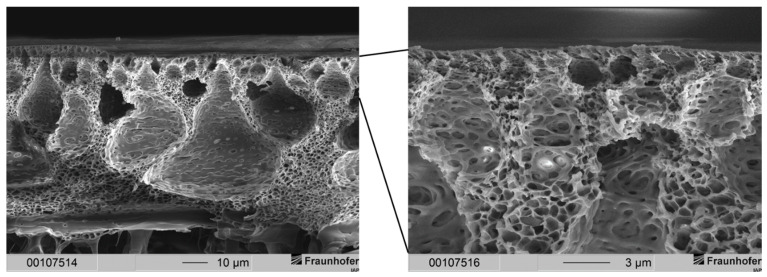
(**Left**) Cross-section of an integrally skinned PSU flat sheet membrane without PDMS protective layer. The PSU concentration was increased to 25 wt.-%. The solvent ratio was kept constant. The precipitation medium used was distilled water. (**Right**) Enlarged view of the top layer with dense separation layer.

**Figure 6 membranes-12-00654-f006:**
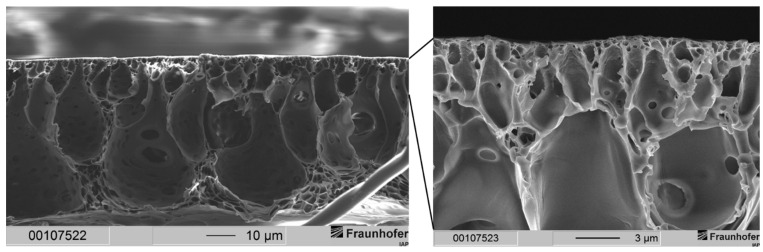
(**Left**) Cross-section of an integrally skinned PSU flat sheet membrane without PDMS protective layer. The PSU concentration was increased to 30 wt.-%. Solvent ration was kept constant. The precipitation medium used was distilled water. (**Right**) Enlarged view of the top layer with dense separation layer and macrovoids.

**Figure 7 membranes-12-00654-f007:**
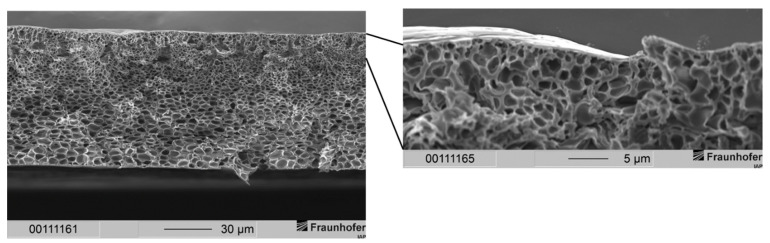
(**Left**) Cross-section of a non-layered PSU flat sheet membrane fabricated according to conditions shown in Table 2. The precipitation medium was distilled water. (**Right**) Enlarged view of the top layer with dense separation layer and sponge-like support structure.

**Figure 8 membranes-12-00654-f008:**
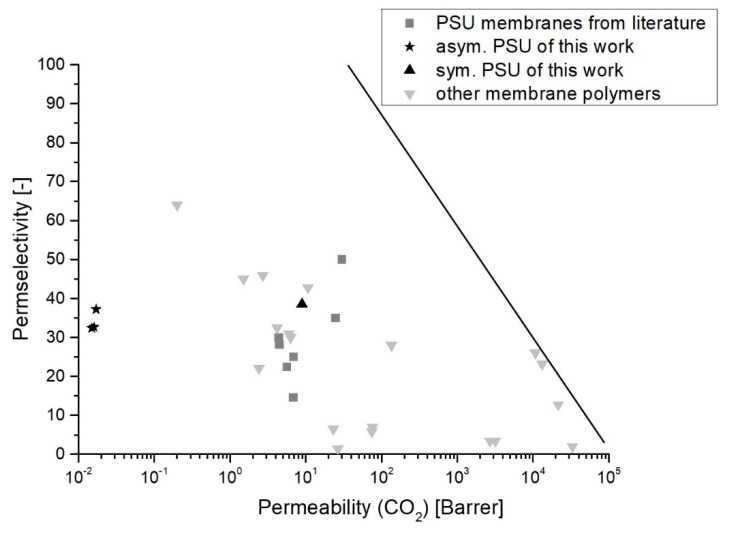
Robeson plot for data from this work and several membrane materials, including PSU membranes from different literature sources [4,15,56,57,58,59,60]. Dark grey squares (■) mark PSU membranes from the literature, black stars (*) mark asymmetric PSU membranes of this work, the black triangle directed upwards (▲) marks the PSU30_EIPS membranes of this work and light grey triangles directed downwards (▼) mark other membrane polymers from literature. The black line marks the Robeson upper bound from 2008 [61].

**Table 1 membranes-12-00654-t001:** Fabrication conditions for PSU flat sheet membranes.

Condition	Value and Unit
composition of casting solution	20 wt.-% PSU 60 wt.-% THF 15 wt.-% DMAc 5 wt.-% MeOH
gap height of casting knife	250 µm
casting speed	17 mm/s
free standing duration	≈3 s
precipitation medium	ethanol for 4 min
washing step	60 s under running water (top and bottom side each)
drying step	16 h in fume hood
lab temperature	22 °C
post-treatment	coating with PDMS protective layer after drying step

**Table 2 membranes-12-00654-t002:** Optimized fabrication conditions for PSU flat sheet membranes.

Condition	Value and Unit
composition of casting solution	25 wt.-% PSU 58.28 wt.-% THF 3.72 wt.-% DMAc 13 wt.-% MeOH
gap height of casting knife	250 µm
casting speed	17 mm/s
free standing duration	≈3 s
precipitation medium	water for 4 min
washing step	60 s under running water (top and bottom side each)
drying step	16 h in fume hood
lab temperature	23 °C
post-treatment	coating with PDMS protective layer after drying step

**Table 4 membranes-12-00654-t004:** Calculated separation layer thickness in asymmetric PSU flat sheet membranes.

Membrane	Separation Layer Thickness (Å)	CO_2_ Permeability Based on New Separation Layer Thickness (Barrer)
PSU25_80:20	230	0.015
PSU30_80:20	384	0.016
PSU25_94:6	463	0.017

**Table 5 membranes-12-00654-t005:** Diffusion coefficients of the gases O_2_, N_2_, CH_4_ and CO_2_ in PSU and PDMS flat sheet membranes.

Membrane	Diffusion Coefficient				References
	O_2_	N_2_	CH_4_	CO_2_	
PSU20_EtOH	1.72 × 10^−8^	3.47 × 10^−9^	2.40 × 10^−9^	1.43 × 10^−8^	this work
PSU25_80:20	2.51 × 10^−14^	5.98 × 10^−15^	3.15 × 10^−15^	1.57 × 10^−14^	this work
PSU30_80:20	3.41 × 10^−14^	9.10 × 10^−15^	5.64 × 10^−15^	2.29 × 10^−14^	this work
PSU25_94:6	3.43 × 10^−14^	7.69 × 10^−15^	4.43 × 10^−15^	2.59 × 10^−14^	this work
PSU30_EIPS	4.06 × 10^−8^	1.07 × 10^−8^	6.24 × 10^−9^	2.46 × 10^−8^	this work
PSU	---	---	4.44 × 10^−9^	4.40 × 10^−8^	[13]
PSU	5.0 × 10^−8^	1.0 × 10^−8^	4.0 × 10^−9^	2.0 × 10^−8^	[54]
PSU	---	---	---	2.60 × 10^−8^	[10]
PDMS	3.40 × 10^−5^	3.40 × 10^−5^	2.20 × 10^−5^	2.20 × 10^−5^	[55]

## Data Availability

Not applicable.

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
