# Peer review of "Tuning the Morphology and Gas Separation Properties of Polysulfone Membranes"

_membranes, 2022, doi:10.3390/membranes12070654_

Round 1

Reviewer 1 Report

The manuscript describes the new tuning formulation for Polysulfone (PSf) Flat sheet membranes for gas separation application. These manuscript is easy to read and by altering/revise several inputs below, this manuscript can be considered for publication in this journal.

1.  unclear novelty: Authors should clearly highlight the main modification in the casting solution in the manuscript.

2. not recent references used: As the PSf membranes fabrication was being research widely and recently, this manuscript used most of the ref. with age of more than 10 years. Although the use of these ref. was to explain/justify the fundamental, more recent ref. should be used to highlight the significant interest of this research.

3. Suggestion: the result would be best presented with the trend graph apposed by Robeson (Park, H. B., Kamcev, J., Robeson, L. M., Elimelech, M., & Freeman, B. D. (2017). Maximizing the right stuff: The trade-off between membrane permeability and selectivity. Science356(6343), eaab0530.) to see the significant impact of this study towards the membrane technology innovation.

Author Response

Dear Reviewer,

a point by point answer to your comments is given in the following table. We tried to introduce all of your comments to our manuscript.

Next to some little mistakes in the manuscript, we also revised the abstract and the conclusion of the manuscript. The last paragraph of the introduction was also revised. This was due to a calculating mistake we found in the manuscript. Therefore, the calculated permeabilities and diffusion coefficients for membranes with separation layers below 100 nm differed several orders of magnitude from the manuscript before revision. We are sorry for not noticing this large issue before and want to apologize for the extra work that may arises from the changes in the manuscript.

The authors want to say that Steven Kluge is the corresponding author for this manuscript.

Additionally, we changed the title of the manuscript. If this is an issue, the title does not need to be changed to the revised version.

Reviewer 2 Report

At present, the manuscript is not yet ready for publication and requires a number of corrections and answers to questions.
Unfortunately, there is no line numbering in the manuscript, which complicates the work. The text of the manuscript contains numerous inaccuracies in terminology.
The purpose of the work is formulated vaguely, hence I recommend the authors to formulate it more precisely. The question arises, how did the authors evaluate the completeness of solvent removal from the membrane? It is not entirely clear what happens when going from 25% solutions to 30% solutions, why did the observed morphology form?

I recommend that authors use one definition to describe the "non-solvent induced phase separation (NIPS)" or "wet-phase inversion" method of membrane formation. At present, the authors operate with one term - method. But in reality, we are talking about technique (a sum of techniques) and a method (method).
"Currently, there is no model available to predict membrane morphology by simulations" - In this case, I recommend that the authors indicate that the authors do not know the models at the moment...
"Based on a polymer dissolved in an organic solvent and water as precipitation medium, membrane morphology can be estimated by solubility parameters of solvents, nonsolvents and polymers" - does not fully agree with the authors. In this case, for example - https://doi.org/10.3390/ma13163495 , the solvent crystallizes in the presence of a precipitant, which makes a great contribution to the forming morphology.
"The polymer solution is displaced accordingly in the direction of penetration by the penetrating water before it is solidified by the solvent exchange" - an unsuccessful proposal, it is desirable to redo it.
"Thus, selectivity increases." I suggest to delete.
"fabricated to a polymer film with the aid of a casting" - in this case, it is better for the authors to remove the "polymer" (the film is made from a solution).
"The starting point for the current research was our publication on fabrication and characterization of polysulfone flat sheet membranes." - Bad beginning of a paragraph.
Fig. 2. "The yellow bar marks the bottom end of the cross section of the membrane." - check the position of the line.
"Since the membrane morphology matches with the morphology of our previous publication, the separation layer showed to large expansion" - I recommend adding a link.
Figure 5. It is not necessary to shift the figures in height. I recommend leveling them.
"which means that the polymer in solution shows a so-called ideal chain" - what do the authors mean by the term ideal chain?
"Permeability is strongly depending on solvents used for preparing the casting solutions." - probably here it is better to talk about a pair of solvent-precipitator vs Permeability?!
"The aim of this study was to tune the morphology of PSU flat sheet membranes. The goal was to lower the thickness of the separation layer of asymmetric PSU membranes without losing the sponge like and mechanically stable support structure these membranes already exhibited." - this part can be removed from the conclusions.

Author Response

Dear Reviewer,

a point by point answer to your comments is given in the following table. We tried to introduce all of your comments to our manuscript.

Next to some little mistakes in the manuscript, we also revised the abstract and the conclusion of the manuscript. The last paragraph of the introduction was also revised. This was due to a calculating mistake we found in the manuscript. Therefore, the calculated permeabilities and diffusion coefficients for membranes with separation layers below 100 nm differed several orders of magnitude from the manuscript before revision. We are sorry for not noticing this large issue before and want to apologize for the extra work that may arises from the changes in the manuscript.

The authors want to say that Steven Kluge is the corresponding author for this manuscript.

Additionally, we changed the title of the manuscript. If this is an issue, the title does not need to be changed to the revised version.

Kind regards,

the authors

Round 2

Reviewer 2 Report

The authors responded to previous questions and comments. However, I would like to draw their attention to the fact that there are a number of contradictory points in the work that need to be identified and eliminated. For example, the authors begin their work with an emphasis on environmental protection, while suggesting the use of complex systems that are very difficult (maybe not even possible) to completely regenerate and use again in the process. The authors miss the point that, regardless of the thickness of the separation layer in it (and in the support layers), the structure may vary, and hence the flow values.
Lines 227, 228. It is desirable to add values ​​and references.
Lines 324, 326. I recommend adding references.
Lines 546-550. "Shishatskii et al. proved decreasing density in polymer membranes with decreasing film thickness" - hence it follows that a more defective (misoriented) structure is formed. Due to which then the permeability decreases. In general, in this work, to understand the results obtained, there are not enough data on density and XRD (X-ray diffraction pattern).
Lines 622, 623. "The aim of this study was to lower the thickness of the separation layer of asymmetric PSU membranes without losing the sponge like and mechanically stable support structure." - I propose to remove the aim of the manuscript from the conclusions.

Author Response

Dear reviewer,

Thank your for your revision. We tried to answer all of your questions and comments. Please find the attached pdf with a table to answer your questions and comments. We also tried to introduce your comments to our manuscript.

Kind regards,

Steven Kluge
